# Exploring Synergistic Effects of Bioprinted Extracellular Vesicles for Skin Regeneration

**DOI:** 10.3390/biomedicines12071605

**Published:** 2024-07-18

**Authors:** Manal Hussein Taghdi, Barathan Muttiah, Alvin Man Lung Chan, Mh Busra Fauzi, Jia Xian Law, Yogeswaran Lokanathan

**Affiliations:** 1Centre for Tissue Engineering and Regenerative Medicine, Faculty of Medicine, Universiti Kebangsaan Malaysia, Cheras, Kuala Lumpur 56000, Malaysia; manalhussein240@gmail.com (M.H.T.); barathanmuttiah@ukm.edu.my (B.M.); fauzibusra@ukm.edu.my (M.B.F.); lawjx@ppukm.ukm.edu.my (J.X.L.); 2Department of Anaesthesia and Intensive Care, Faculty of Medical Technology, University of Tripoli, Tripoli P.O. Box 13932, Libya; 3My Cytohealth Sdn. Bhd., Kuala Lumpur 56000, Malaysia; alvinchan@cytoholdings.com

**Keywords:** extracellular vesicles, 3D bioprinting, biogenesis, wound healing, growth factors, skin regeneration

## Abstract

Regenerative medicine represents a paradigm shift in healthcare, aiming to restore tissue and organ function through innovative therapeutic strategies. Among these, bioprinting and extracellular vesicles (EVs) have emerged as promising techniques for tissue rejuvenation. EVs are small lipid membrane particles secreted by cells, known for their role as potent mediators of intercellular communication through the exchange of proteins, genetic material, and other biological components. The integration of 3D bioprinting technology with EVs offers a novel approach to tissue engineering, enabling the precise deposition of EV-loaded bioinks to construct complex three-dimensional (3D) tissue architectures. Unlike traditional cell-based approaches, bioprinted EVs eliminate the need for live cells, thereby mitigating regulatory and financial obstacles associated with cell therapy. By leveraging the synergistic effects of EVs and bioprinting, researchers aim to enhance the therapeutic outcomes of skin regeneration while addressing current limitations in conventional treatments. This review explores the evolving landscape of bioprinted EVs as a transformative approach for skin regeneration. Furthermore, it discusses the challenges and future directions in harnessing this innovative therapy for clinical applications, emphasizing the need for interdisciplinary collaboration and continued scientific inquiry to unlock its full therapeutic potential.

## 1. Introduction

Skin injuries have a severely detrimental impact on the world’s healthcare systems and economies. Acute and chronic wounds affect roughly a billion people worldwide, according to a new survey [1,2]. By 2027, the market for advanced wound care will be valued at MYR 18.7 billion, expanding at a compound annual growth rate (CAGR) of 6.6% from 2020 to 2027 [3]. With a CAGR of 10%, China is anticipated to develop its market to MYR 4 billion by 2027. Over the 2020–2027 timeframe, Japan and Canada are anticipated to experience notable growths of 3.6% and 5.8%, respectively. The United States, Canada, Japan, China, and Europe are anticipated to be the main drivers of a projected 7.2% CAGR in the worldwide antimicrobial dressings market. After the analysis period, it is expected that the collaborative efforts across these geographic regions will augment the market size by MYR 1.8 billion. China is poised to lead in growth rate, with a projected compound annual growth rate (CAGR). Australia, India, and South Korea are expected to lead growth in the Asia Pacific region to MYR 2.6 billion by 2027 [4,5]. Diabetic foot ulcers and wound surgeries are extremely expensive to treat. Although treatments for acute wounds have been developed, choices for chronic wounds are insufficient, and skin grafts are still employed.

The emergence of live cell printing, or 3D bioprinting, represents a revolutionary advancement in tissue engineering and regenerative medicine and encompasses diverse strategies aimed at addressing deformities and traumatic injuries by harnessing the body’s innate capacity for repair and regeneration [6]. This technology enables the precise spatial arrangement of living cells and biologics to fabricate functionalized tissues, layer by layer, using computer-aided techniques [7,8,9]. Amidst a global demand for replacement organs and tissues, 3D bioprinting holds promise in addressing the shortage of donors by manufacturing new functional organs derived from autologous extracellular matrices [10]. Moreover, it offers avenues for improved methodologies in studying diseases within both in vitro and in vivo models, as well as overcoming challenges in drug delivery. 3D bioprinting encompasses two main approaches using cells or employing cell-free techniques. Bioprinting with cells involves the precise deposition of live cells to create tissue-like structures to mimic the native cellular environment. This method enables the recreation of complex tissue architectures and promotes cellular interactions crucial for tissue regeneration. Bioprinted constructs containing cells have the potential for better integration with host tissues, facilitating tissue repair and regeneration. However, challenges such as nutrient supply limitations, immunological rejection, and poor post-implantation engraftment hinder the seamless transition to clinical application. Regulatory complexities and financial burdens associated with cell culture and storage further impede progress. Conversely, cell-free bioprinting utilizes biomaterials, such as hydrogels or decellularized extracellular matrices, to provide a supportive scaffold for tissue engineering. Cell-free bioprinting streamlines the fabrication process, eliminating the need for cell culture and maintenance, which enhances scalability and reduces costs [11]. While cell-free bioprinted constructs may lack some aspects of cellular functionality, they offer advantages in terms of reduced immunogenicity and simplified fabrication processes, making them attractive for various tissue engineering applications. However, further research is needed to enhance the biological functionality and tissue integration of cell-free bioprinted constructs for optimal clinical outcomes. Recently, alternative strategies, such as cell-free bioinks incorporating bioactive molecules such as growth factors (GFs) or extracellular vesicles (EVs), offer promising avenues to circumvent these obstacles, providing scalable and cost-effective solutions for regenerative medicine. These approaches span from the transplantation of stem cells or biological molecules to the replacement of organs or tissues with engineered cellular structures grown ex vivo. Studies have demonstrated their efficacy in promoting tissue regeneration in various preclinical models, including those of cardiovascular disease, neurodegenerative disorders, and musculoskeletal injuries [11]. Moreover, EV-based therapies hold promise for addressing the challenges associated with traditional cell-based approaches, such as immune rejection and tumorigenicity [12].

By leveraging these innovative strategies, researchers aim to overcome the limitations of traditional regenerative therapies and pave the way for more effective and accessible treatments for a wide range of conditions. The integration of bioactive biomaterials into regenerative medicine not only enhances therapeutic outcomes but also offers new avenues for personalized and minimally invasive interventions. As research in this field continues to evolve, it holds the promise of revolutionizing the treatment landscape for deformities and traumatic injuries, ultimately improving patient outcomes and quality of life.

## 2. Extracellular Vesicles (EVs)

EVs are microscopic, membrane-enclosed structures released by cells. These miniature messengers play a critical role in intercellular communication across diverse life forms, from single-celled archaea to complex eukaryotes. Notably, EVs extend communication beyond the confines of an individual organism, facilitating interactions between different species and kingdoms. Over the past two decades, research has revealed EVs as key players in various physiological and pathological processes [13]. This stems from their ability to transmit signals, influencing not just neighboring cells but also distant ones. EVs exhibit remarkable diversity, categorized into distinct subtypes based on their origin, size, and molecular cargo. Exosomes, ranging from 30 to 150 nanometers, are formed within the cellular endosomal pathway. Ectosomes, which are larger at 100–1000 nanometers, bud directly from the cell’s outer membrane. Apoptotic bodies, shed by dying cells or left behind by migrating cells, represent additional EV subtypes with sizes varying from 50 to 5000 nanometers (see Figure 1). Notably, amoeboid cancer cells can release even larger oncosomes, reaching sizes between 1000 and 10,000 nanometers. Even smaller entities, exomeres and supermeres (less than 50 nanometers), are classified as extracellular particles (EPs), with their formation mechanisms still under investigation. EVs act as cellular mail carriers, encapsulating a diverse cargo of proteins, lipids, nucleic acids, and metabolites for delivery to recipient cells. The specific cargo composition within an EV dynamically reflects the health state of the originating cell. This characteristic offers the potential for an “EV fingerprint”, a unique molecular signature indicative of the tissue source, cell type, or even the presence of a specific disease.

Tetraspanins (CD9, CD81, and CD63), lipids, integrins, the major histocompatibility complex (MHC), heat shock proteins (HSPs), growth factors, circular RNAs (circRNAs), microRNAs, mRNA, proteins, long non-coding RNAs, and genomic DNA are among the crucial biological molecules transported by extracellular vesicles (EVs) from their host cells [14,15,16]. Due to their diverse composition, EVs are considered ideal candidates for both diagnostic and therapeutic applications. However, for effective EV therapy, it is essential to properly isolate and characterize EVs [17,18]. Isolation techniques ensure purity by removing impurities that could compromise the effectiveness of treatment, while characterization reveals the EV composition and functional properties, enabling tailored delivery. Currently, there are no universally standardized procedures for isolating EVs. Although repeated centrifugation is a commonly used method to extract apoptotic bodies and microvesicles including small EVs (sEVs), alternative methods such as ultrafiltration, precipitating agents (e.g., polyethylene glycol), microfluidics, immunoaffinity capture, and size-exclusion chromatography (SEC) have also demonstrated effectiveness in isolating EVs [17,19]. When isolating EVs from a conditioned medium (CM) in vitro, several variables must be considered, including the primary cell source (donor characteristics), passage number, CM volume/change frequency, composition (e.g., EVs-depleted FBS or FBS-free), and CM harvesting conditions. The isolation methodology for cell-derived EVs is generally similar to that for oral fluid [20,21]. Following the removal of the CM from the cell culture and centrifugation to remove cell debris, apoptotic bodies can be pelleted by centrifugation at 2000 g for 15 min, followed by obtaining microvesicles at 20,000 g for 20 min. Finally, sEVs can be enriched using a size exclusion chromatography column or ultracentrifugation (>100,000× *g* for >1 h). Pure sEVs with a cup-shaped morphology can be identified using transmission electron microscopy (TEM). The characterization of EVs should include an assessment of a cup-shaped morphology, EV-enriched protein analysis, and EV size distribution, as recommended by the minimal information for studies of extracellular vesicles (MISEV) [22]. Techniques such as TEM, nanoparticle tracking analysis (NTA), dynamic light scattering (DLS), the enzyme-linked immunosorbent assay (ELISA), Western blot (WB) analysis, and nanoscale flow cytometry can be employed for EV characterization post-isolation [23]. Despite that, the challenge of standardizing nomenclature for each EV subtype remains.

## 3. Skin Regeneration

### 3.1. Skin

The epidermis, dermis, and subcutaneous tissue make up the three layers of the skin, which is the biggest organ in the body (Figure 2). It serves as a barrier against physical (such as external forces, dryness, UV radiation, and temperature fluctuations), chemical, and pathological disturbances (such as infection). It is the outermost layer of the body. From the inside to the outside, it is continuously replenished, and the immune system benefits from the replacement of cells. The skin has multiple appendages; sweat glands are crucial for controlling body temperature, and hair functions as a tactile organ by combining with numerous nerves [24].

Derived from the ectoderm, the epidermis is a stratified squamous epithelium. Melanocytes are mostly found in the basal layer of the epidermis, which is separated into five layers: the basal layer, light layer, granular layer, spinous layer, and stratum corneum. During differentiation, keratinocytes that proliferate and divide in the basal layer migrate to the top layers, causing keratinization, which includes enucleation and shedding from the stratum corneum’s surface. Strong intercellular connections between keratinocytes and skin turnover create a barrier that stops the body from losing water and allowing external objects to penetrate inside. Furthermore, the epidermal basement membrane closely binds to the epidermis and dermis, and controls the movement of substances across this barrier. It is mostly made of type IV collagen and laminin, which are produced by the cells that make up the basement layer. The connective tissues of mesenchymal origin include the dermis and hypodermis. The hypodermis is a sparse connective tissue that contains a large number of fat cells, while the dermis is a thick, dense connective tissue with dense collagen fibers. Rough collagen fibers make up the reticular layer, which makes up the majority of the dermis. Elastic fibers, on the other hand, form a delicate meshwork on the dermis’ surface that leads to the basement membrane. Fibroblasts make up the bulk of the cells in the dermis, although there are also migratory cells like lymphocytes and macrophages. Collagen and elastic fibers in the hypodermis are orientated in different directions: vertically in the skin’s less mobile parts, including the palms of the hands and the head, and horizontally in the skin’s highly mobile areas. A network of thin arteries on the surface of the hypodermis branches out into the dermis and subcutaneous tissue, while thinner arteries themselves reach the dermis’ surface. The upper dermis and superficial layers of the hypodermis include a network of veins that are involved in controlling body temperature [25].

### 3.2. Wound Healing

A wound represents a disruption in the anatomical structure and function [26], often resulting from surgical procedures, traumatic incidents, or burns. Typically, acute wounds can heal within 8 to 12 weeks owing to the remarkable regenerative abilities of the skin [26,27]. However, in cases of deep or extensive wounds that penetrate the dermis layer and involve persistent inflammation, such as burns, bedsores, or diabetic ulcers, the healing process may be significantly delayed [28]. The intricate process of wound healing requires a favorable microenvironment and precise coordination among multiple cell types (Figure 3). Acute skin wounds progress through four distinct stages: hemostasis, inflammation, proliferation, and maturation. Following an injury, platelets aggregate to form a blood clot, initiating the hemostatic response [2,26]. Subsequently, the inflammatory phase ensues, characterized by the accumulation of blood and inflammatory cells, including leukocytes, macrophages, and platelets, at the wound site [29]. Macrophages release proteolytic enzymes to remove foreign materials and facilitate tissue cleaning, while inflammatory cells phagocytose pathogens. Growth factors play a crucial role in regulating cell proliferation, and an imbalance in their levels can perpetuate the inflammatory phase, impeding the normal healing process. After inflammation subsides, the proliferative phase begins, marked by the formation of granulation tissue. Fibroblasts and endothelial cells proliferate and migrate to the wound site in response to growth factors released during the inflammatory phase, contributing to the formation of granulation tissue. This tissue fills tissue defects, protects the wound surface from infection, and promotes blood flow and cell growth [30]. The maturation phase, which may extend over several years, represents the final stage of wound healing. During this phase, dense scar tissue forms as capillaries regress, and the granulation tissue undergoes remodeling. Collagen fiber maturation occurs, and over time, the scar tissue approaches the strength of normal skin and becomes less noticeable. A failure to complete the sequence of wound-healing mechanisms can result in chronic wounds, characterized by prolonged inflammation and impaired cellular activity [31,32]. To enhance wound healing in chronic skin injuries, barriers such as persistent inflammation, ischemia-reperfusion-induced free radical formation, and infection must be addressed. Moreover, both local and systemic factors, including aging, hypoxia, poor circulation, recurrent wound damage, and nutritional deficiencies, can influence the wound healing process and need to be managed effectively.

### 3.3. Role of EVs from Different Cellular Origins for Wound Healing

In the context of regenerative medicine, considerable interest has been taken in extracellular vesicles (EVs) because of their therapeutic potential. Originating from various sources, these vesicles have shown promising effects on tissue regeneration, particularly in wound healing; since they carry bioactive molecules like growth factors and signaling proteins, they also support important cellular processes in wound healing and tissue repair. Furthermore, when incorporated into sophisticated 3D printed bandages, these extracellular vesicles provide a precise and regulated method of delivery, improving their effectiveness in stimulating tissue growth. This new method shows potential in tackling issues linked to traditional wound healing methods, presenting opportunities to enhance patient results in conditions such as diabetic foot ulcers and other persistent wounds. Based on Table 1 below, the role of MSC-determined EVs during the time spent recuperating from injury is detailed. Mesenchymal stem cells (MSCs) have the ability to change into various cell types and discharge extracellular vesicles (EVs) with significant variables for tissue recuperation. These EVs control inflammation by favoring M2 macrophage polarization to enable angiogenesis and epithelial recovery. Moreover, they assume a part in changing tissues by controlling collagen creation and the improvement of scars [33]. Extracellular vesicles form macrophages to mend wounds. Additionally, EVs isolated from macrophages, containing cytokines and miRNAs, decrease inflammation and improve wound healing by supporting angiogenesis and re-epithelialization. Additionally, they influence the arrangement of scars by controlling the creation of collagen [34]. Cell-inferred extracellular vesicles (EVs) from the skin recuperate wounds. Skin-related cells like dermal fibroblasts and keratinocytes improve the wound healing process by advancing cell development, the arrangement of fresh blood vessels, and the creation of collagen. They aid in the renovating of tissues and the constriction of wounds, significant for compelling mending [11]. EVs acquired from blood also aid in mending wounds. Blood-inferred extracellular vesicles (EVs) from platelet-rich plasma (PRP) and umbilical string blood (UCB) plasma advance injury recuperation by further developing cell expansion, relocation, and tissue recovery. They speed up the course of tissue recovery by advancing re-epithelialization, angiogenesis, and collagen synthesis [35]. Similarly, EVs from endothelial cells have also contributed to wound healing. EVs from endothelial cells assume a part in controlling injury-mending components that affect the activities of other endothelial cells [36]. For example, EVs assist by expanding microvascular thickness and collagen production, while others can thwart the cycle by diminishing angiogenesis and the collagen framework.

### 3.4. Role of GF in EVs for Wound Healing

EVs express their functions through unique biomolecules, which include proteins, lipids, nucleic acids, and others. These signaling molecules come in various forms as described in Table 2, including PDGF, FGFs, TGFs, and VEGF [37]. GFs like PDGFs, FGFs, EGF, TGF-β, VEGF, and GM-CSF play pivotal roles in wound healing, particularly in burn-related chronic wounds where deficiencies in these factors are observed [38]. Among them, Fibroblast Growth Factors (FGFs), notably FGF-2, FGF-7 (or KGF-1), and FGF-10 (or KGF-2), regulate fibroblast cell migration, angiogenesis, and wound repair signaling pathways, promoting tissue regeneration. Basic Fibroblast Growth Factor (bFGF) emerges as a potent mitogenic factor crucial for wound repair, angiogenesis, and tissue granulation [39]. Epidermal Growth Factor (EGF) stimulates keratinocyte migration, fibroblast function, and granulation tissue formation, enhancing wound healing rates, especially in chronic wounds. Transforming Growth Factor-Beta (TGF-β) family members, particularly TGF-β1, regulate mesenchymal cell functions, ECM production, and remodeling during wound healing, while the Vascular Endothelial Growth Factor (VEGF) family promotes angiogenesis and endothelial cell proliferation, critical for blood vessel formation [40].

The IGF family, particularly IGF-1, mends wounds and empowers tissue fixing by empowering fibroblast and keratinocyte movement and improvement. Keratinocyte expansion component (KGF) advances keratinocyte relocation and multiplication, which supports tissue recovery and reepithelization. It additionally gives imminent treatment choices to constant injuries. Granulocyte-Macrophage Settlement Invigorating Element (GM-CSF) controls incendiary responses and cell movement to animate the development of fresh blood vessels and rush the mending of wounds. Granulation tissue arrangement and wound mending are subject to PDGF family proteins, for example, PDGF-AA and PDGF-BB, to advance fibroblast expansion, ECM blend, and myofibroblast separation. As indicated by Garoufalia et al. (2021), treatments in view of PDGF advance tissue fixing and conclusion, which thusly speeds up the mending of wounds [41].

## 4. 3D Bioprinting

3D bioprinting is a revolutionary technology that allows for the precise deposition of bioinks containing cells, growth factors, and other bioactive substances to produce tissue-like structures resembling in vivo tissue characteristics [13]. By controlling bioink deposition, interconnected holes are created in layer-by-layer constructions, facilitating the infusion of nutrients, gases, and cellular communication. This method can tailor scaffolds for various tissue engineering applications, achieving the desired shape, size, porosity, and interconnectivity [42,43,44]. To achieve optimal results, bioprinting techniques must support cell viability and structure, accommodate various bioink viscosities and crosslinking groups, and enable precise spatial arrangement control over clinically significant dimensions. Bioinks, composed of biomaterials, living cells, and biomolecules, present a significant challenge, requiring properties that meet specific tissue construct needs [45,46]. Hydrogel-based bioinks, crucial for cell viability and stress protection during production, are commonly used due to their high water content [47,48]. Viscosity, gelation, rheological characteristics, and crosslinking abilities are key properties to assess before printing properties [49,50]. Bioinks can consist of synthetic or natural materials, with hydrogels derived from substances like collagen, fibrin, hyaluronic acid, and alginate [51,52]. Recent advances include incorporating growth factors and ECM proteins to enhance cell behavior and tissue regeneration. Printability, influenced by bioink viscoelasticity and mechanical strength, is critical for successful bioprinting [53,54]. Common bioprinting methods include inkjet, extrusion, and laser-based technologies, each with unique advantages and limitations [55,56]. Laser-based bioprinting offers precision but is limited by the availability of biocompatible resins. Inkjet printing enables material-saving deposition but requires low-viscosity bioinks. Extrusion printing is versatile, affordable, and capable of depositing high cell densities, but suffers from lower printing resolution. Despite challenges, 3D bioprinting holds immense potential for advancing tissue engineering and regenerative medicine [57].

### 4.1. Uses of 3D Bioprinting for Skin Regeneration

Although applications for 3D bioprinted EVs in skin regeneration have developed during the past few years, very few investigations have been carried out. The technology of 3D printing has advanced into an effective manufacturing method that is widely employed due to its benefits over conventional techniques, including rapid prototyping and end-user customization. This technology finds application in a variety of fields, including the biomedical field. It does, in fact, constitute a useful tool for the realization of biodevices (drug delivery systems, microfluidic bioreactors, and biosensors) [58]. Bioprinting provides the precise manipulation of bioink placement, facilitating the development of intricate personalized designs with great precision. The accuracy at this level is essential for tissue engineering applications, which need precise architectures to replicate natural tissues [59]. Additionally, bioprinting allows for the integration of various materials or cells into the printed design in a step-by-step manner. This ability enables the development of diverse tissue structures with precise control over cell placement, a task that is difficult to accomplish using conventional techniques [60]. On the other hand, bioprinting allows for the placement of bioinks in a three-dimensional (3D) environment, making it possible to create complex 3D structures that mimic the natural tissue layout more accurately. This is especially crucial in fields like organ printing, which aims to replicate the intricate shape and capabilities of real organs [61]. In general, bioprinting is a versatile and accurate method for creating tissue structures, which makes it a valuable asset in tissue engineering and regenerative medicine [62].

Combining the regenerative potential of both bioprinting and EVs, this perspective examines the existing literature on 3D bioprinted EVs in tissue engineering, focusing on angiogenesis, osteogenesis, chondrogenesis, myogenesis, and carcinoma prevention. It discusses technical hurdles and future directions for 3D bioprinted EVs in biofabrication and tissue engineering, proposing a personalized bioprinted EV concept and workflow for clinical translation studies [63]. For downstream applications, it is well-recognized that cell source, EV enhancement, and characterization are essential. The cell source, conditioned media collection conditions, current techniques for isolating and characterizing EVs, as well as a thorough application for each chosen study, as reviewed by [51] for 3D bioprinting, includes drug screening, tissue engineering, and in vitro disease models. Several 3D bioprinting techniques have been successfully used to create a variety of tissue constructs that imitate biological tissue and organs, including bone, vascular, skin, cartilage, and neural structures [8,56,64]. For instance, 3D bioprinting technology for bioengineered skin tissue is becoming more and more significant as evidenced by the treatments that are presently available on the market. The continuous pores of the scaffolds are effective for ECM deposition, oxygen and nutrient exchange, and the infiltration of cells and microvessels from the surrounding tissue, in addition to providing physical protection through the spatial complementation of skin defects [65]. The numerous uses of 3D bioprinting for tissue regeneration have recently been thoroughly reviewed [5,57,66,67,68,69,70], particularly in the fields of skin, muscle, cardiac, and orthopedic tissue regeneration. These extensive investigations have shed important light on the developments and prospects of 3D bioprinting methods to speed up the regeneration of these particular tissues. Different reviews have illuminated the encouraging therapeutic outcomes and prospects for different tissue types by examining the novel techniques and nanomaterials used in bioprinting, as well as the incorporation of biologically relevant cells, growth factors, and other biomolecules. To supply all the necessary qualities for each target tissue, more research is necessary.

In Table 3, several recent developments of incorporating EVs from different cellular types into bioinks were reviewed. In 2021, Bari and colleagues utilized poly(ε-caprolactone) (PCL) and a freeze-dried lyosecretome from MSC for bone regeneration [71]. PCL is a synthetic biodegradable polymer favored for its excellent biocompatibility, slow degradation rate, and mechanical properties. Thus, it is suitable for creating scaffolds that support areas like bone, cartilage, and soft tissue engineering. Their scaffold demonstrated the controlled release of the lyosecretome, promoting bone healing and regeneration in the preclinical model. Ferroni and colleagues developed an advanced 3D wound dressing using hyaluronic acid derivatives (HAs) and small extracellular vesicles (sEVs) from human mesenchymal stem cells (MSCs) [72]. While conventional wound dressings like hydrogels have benefits, they need improvements for mechanical properties and residence time [73]. HA, due to its similarity to native ECM and involvement in tissue repair, is ideal for this purpose. 3D printing allows personalized, bioactive dressings to be made, and recent studies have shown success using hydrogels to deliver sEVs for wound healing. In this study, MSC-sEV production was optimized, and MeHA-based dressings were created and loaded with MSC-sEVs. Testing in a diabetic mouse model showed improved healing outcomes, indicating the potential of this approach for diabetic foot ulcer management [72]. On the other hand, a study has provided insights into optimizing the composition of a silk–alginate (SA-SF) hydrogel, the degumming time, and the crosslinking method; researchers were able to achieve improved shape fidelity, mechanical properties, and the controlled release of bioactive molecules like EVs. These findings pave the way for developing SA-SF bioinks with tunable mechanical and EV-release properties for scaffold 3D printing in TE [74]. Similarly, Bar et al. (2022) have also developed a cardiac patch composed of alginate sulfate (AlgS) and cardiac stem cell-derived EVs [75]. Their model strategically demonstrated the repair of damaged cardiovascular tissue, improving outcomes and survivability in an animal model. Bar and colleagues have also successfully bioprinted using alginate (LVG) and miR-199a-3p-enhanced EVs from THP-1-derived activated macrophages. This model had more focus on preserving the qualities of CP via the sustained release of the EVs and to preserve the viability of cardiomyocytes [76]. Born et al. integrated MSC EVs into GelMA bioinks via 3D printing, mitigating rapid clearance. By adjusting the crosslinker concentration, they ensure sustained release, maintaining bioactivity. This method, which has been shown to be effective in promoting angiogenesis, offers controlled EV delivery. It holds potential for diverse therapeutic applications, notably in wound healing [77]. Furthermore, Maiullari et al. (2021) also demonstrated a biocompatible construct made of a GelMA bioink incorporated with HUVEC cell-derived EVs [78]. The 3D bioprinted construct loaded with endothelial-derived EVs successfully induced organized neovascularization, enhancing blood vessel formation and tissue regeneration when subcutaneously implanted in vivo.

### 4.2. Combining Imaging Techniques and AI with 3D Bioprinting

In the context of EVs, bioprinting technology possesses several benefits over traditional methods. As shown in Figure 4, 3D bioprinted EVs can be manifested using different GF-enriched EVs acquired from biopsied cells or their recombinant alternatives. This results in the heterogenous properties found in native tissue constructs and a near-homologous graft. Unlike the syringe method, bioprinting offers high spatial resolution for the precise three-dimensional placement of the materials, essentially fabricating the constructs to accurately mimic the intricate architecture of complex tissues [11,44,51]. Furthermore, the fixed EV types at each layer should signal and recruit the corresponding cells to mend the wound and assimilate with the implanted structure. This construct also overcomes the lack of mechanical strength, simultaneously increasing the retention rate of EVs and maximizing their efficacy. Comparably, the automation of bioprinting not only ensures consistent and reproducible results, minimizing human errors, but also presents as a scalable option for the mass production of the tissue constructs [79]. Since it is an acellular alternative, EVs are easily manipulated with less concern for their viability or sensitivity to changes over time. For those reasons, bioprinting has manufacturing and commercial advantages over manual, labor-intensive methods. The imaging technology, coupled with advancements in artificial intelligence (AI), is well regarded to revolutionize wound management [79]. Modern imaging techniques including magnetic resonance imaging (MRI), computerized tomography (CT) scans, or ultrasounds provide detailed imaging data on wound topology and tissue composition [80]. AI techniques could run this data through an algorithm, providing a swift assessment of wound size, depth, prognosis, and/or potential complications [81,82]. As such, it is more efficient at producing 3D-bioprinted EVs, which are limited by time-consuming data analysis and complex computational processing. Future AI development may enhance personalized constructs by integrating patient-specific data, including genetic information and previous medical history, or by selecting and dosing EVs for optimal wound management. Ultimately, this helps to minimize chronic wound complications, improving patient welfare and overcoming healthcare costs.

## 5. Future Direction and Conclusions

Future directions in tissue engineering and regenerative medicine involve advancing bioink development for improved cell viability and differentiation, enhancing the therapeutic efficacy of extracellular vesicles through surface modification and cargo loading, refining bioprinting techniques for higher resolution and organ-on-a-chip platforms, integrating bioreactor systems to mimic physiological environments, addressing regulatory challenges for clinical translation, embracing personalized medicine approaches, and establishing biofabrication standards and guidelines to ensure reproducibility and safety. To summarize, the mix of 3D bioprinting, EVs, and their development factors (GFs) shows extraordinary potential in the field of tissue designing and regenerative medication. These trend setting innovations present novel opportunities to conquer the downsides of customary medicines and change the manner in which conditions like skin recovery and wound mending are dealt with. The exact testimony of bioinks containing cells or EVs using 3D bioprinting considers the formation of complex tissue structures that imitate normal tissues. 3D bioprinted techniques can possibly uphold tissue recovery and useful recuperation by replicating regular tissue structures and making a reasonable microenvironment for cell connections. EVs, containing different bioactive particles, have a significant capability in transmitting among cells and keeping up with tissue balance. Adding EVs to 3D printed organic designs builds their capacity to recuperate by further developing correspondence between cells, controlling aggravation, and supporting the development of fresh blood vessels and the rebuilding of tissue. Also, EV-based therapies have benefits like diminished safe reactions and smoothed out assembling techniques, which make them engaging for use in clinical settings. EVs are fundamental for controlling cell conduct and tissue fixing processes. 3D bioprinted techniques can work on the regenerative cycle, invigorate vein arrangement, and accelerate wound recovery by conveying development variables to the injury site through outer applications or inward enactments. 3D bioprinting takes into consideration the precise administration of EV conveyance, prompting designated and controlled discharge for ideal restorative outcomes. As a rule, in the context of the cooperative mix of 3D bioprinting, EVs show extraordinary potential for advancing tissue designing and regenerative medication. Specialists can upgrade patient results and personal satisfaction by handling specialized deterrents and amplifying the abilities of these innovations to make custom-made and proficient medicines for different circumstances.

## Figures and Tables

**Figure 1 biomedicines-12-01605-f001:**
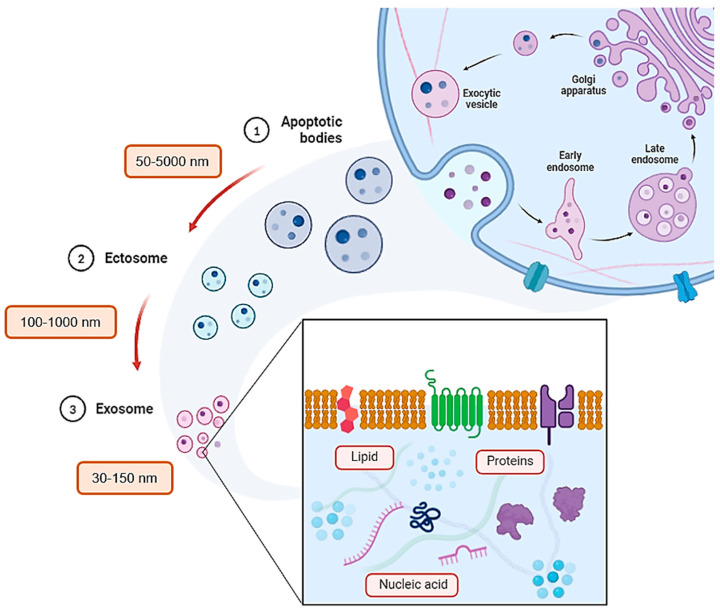
Schematic illustration of intracellular manifestation of the extracellular vesicles (EVs), followed by its categorical size (nm) and its supposed content.

**Figure 2 biomedicines-12-01605-f002:**
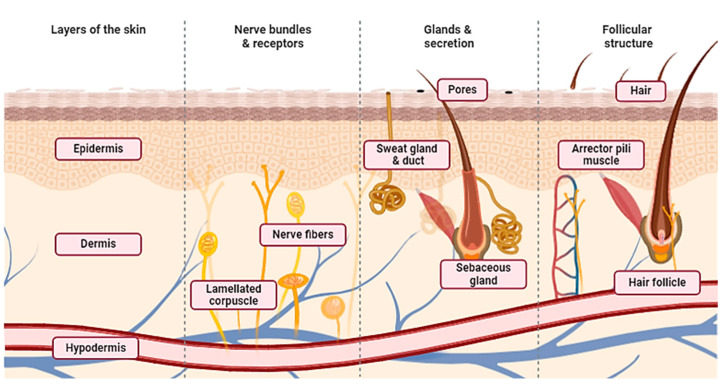
Schematic illustration of the human skin and its supporting tissues.

**Figure 3 biomedicines-12-01605-f003:**
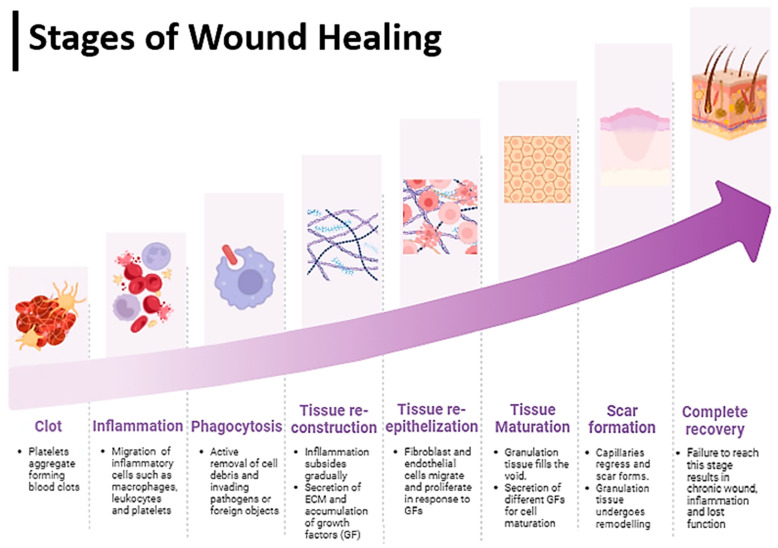
Schematic illustration of the wound healing process.

**Figure 4 biomedicines-12-01605-f004:**
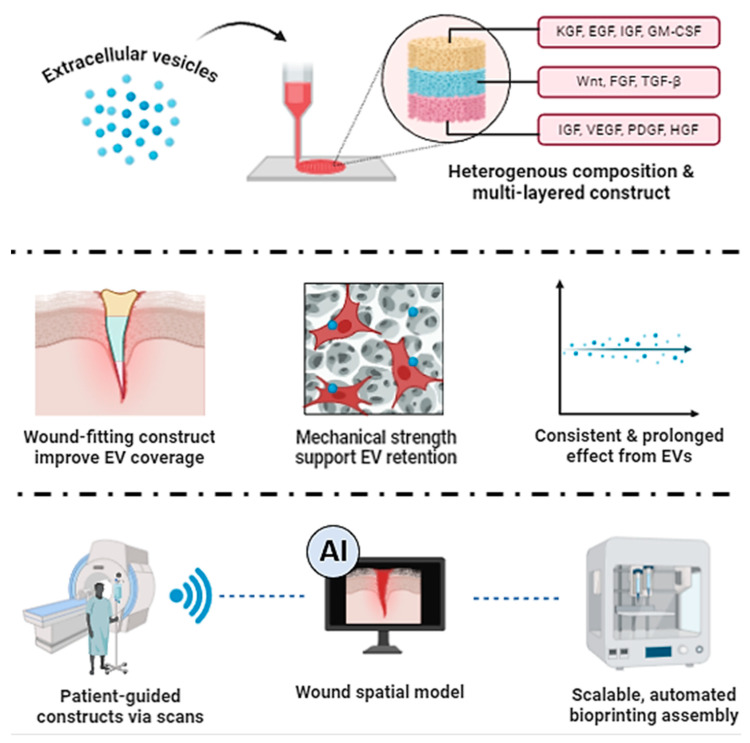
Schematic illustration of 3D bioprinted EVs using different GF-enriched or tissue-specific EVs, their wound management, and the potential of imaging and AI-assisted technology.

**Table 1 biomedicines-12-01605-t001:** Role of different EVs by source in wound healing process.

Source	Origin	Role in Wound Healing
[33]	MSCs	Induce anti-inflammatory response in macrophages, promote angiogenesis, support epithelial recovery, regulate collagen production, and scar formation.
[34]	Macrophages	Reduce inflammation, enhance wound healing through angiogenesis and re-epithelialization, and modulate collagen synthesis to regulate scar formation.
[10]	Skin-related cells	Promote cell growth, angiogenesis, collagen synthesis, tissue remodeling, and wound contraction to facilitate effective wound healing.
[35]	Blood-derived	Enhance cell proliferation, migration, and tissue regeneration, expedite re-epithelialization and angiogenesis, and promote collagen synthesis to accelerate wound healing.
[36]	Endothelial cells	Influence endothelial cell activities, enhance microvascular density, regulate collagen deposition, and modulate angiogenesis, thereby impacting wound healing positively or negatively.

**Table 2 biomedicines-12-01605-t002:** Role of GFs in wound healing.

Growth Factors	Function
FGF Family (FGF-2, FGF-7, FGF-10)	Regulates fibroblast cell migration, angiogenesis, and wound repair signaling pathways
EGF	Stimulates keratinocyte migration, fibroblast function, and granulation tissue formation
TGF-β Family (TGF-β1)	Regulates mesenchymal cell functions, ECM production, and remodeling during wound healing
VEGF Family	Promotes angiogenesis and endothelial cell proliferation
IGF Family (IGF-1)	Mends wounds and empowers tissue fixing by empowering fibroblast and keratinocyte movement and improvement
KGF	Advances keratinocyte relocation and multiplication, which supports tissue recovery and reepithelization
GM-CSF	Controls inflammatory responses and cell movement to animate the development of fresh blood vessels and rush the mending of wounds
PDGF Family (PDGF-AA, PDGF-BB)	Advances fibroblast expansion, ECM blend, and myofibroblast differentiation

**Table 3 biomedicines-12-01605-t003:** The 3D bioink wound dressings using different materials.

Study	Material	Improvement
[71]	Poly(ε-caprolactone) (PCL) and freeze-dried lyosecretome from MSC.	Homogeneous loading of protein and EVs and a controlled slow release.
[78]	GelMA bioink with EV from HUVEC cells.	The implant demonstrated in situ retention and formation of functional vasculature.
[75]	A cardiac patch composed of alginate sulfate (AlgS) and EVs.	Superior integration and sustained release of EVs. Better assimilation of patch into cardiac tissue.
[77]	GelMA bioinks with MSC EVs.	Sustained release of bioactive EVs and promoting angiogenesis.
[74]	Silk–alginate (SA-SF) hydrogel.	Improved shape fidelity, mechanical properties, and controlled release of bioactive molecules.
[72]	Hyaluronic acid derivatives (HA) and small EVs from human MSC.	Improved mechanical properties and residence time for wound dressings.
[76]	Alginate (LVG) and THP-1-derived activated macrophages.	Inclusion of EVs yielded superior cell viability and lower ratio of apoptotic CM.

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
