# Peer review of "Exploring Synergistic Effects of Bioprinted Extracellular Vesicles for Skin Regeneration"

_biomedicines, 2024, doi:10.3390/biomedicines12071605_

Round 1

Reviewer 1 Report

Comments and Suggestions for Authors

Section 3 about growth factors is not really needed, as they aren't the only cargo that EVs transport. What about lipids, DNA, RNA..? 

I do not understand the reasoning behind including GF in the review as they can be inside the EV or as a synergic effect. Are there published papers where they combined all of them? Aren't EVs supposed to have an equivalent effect or contain this GF? What would be the complications associated with including EVs and multiple GF, and how the dosing be calculated? 

The images from the figures are not of very good quality (e.g. Figures 3, 4). The authors should include more figures where results from the published papers appear not only resume 3-5 papers in tables 1 and 3. The selection of papers is based on what consideration? Only 3 or 5 papers were published in the field in the past years? 

The tables are not discussed in the text, so I recommend making tables with complete information or changing them by figures where results from papers are presented. 

Author Response

Comments 1: Section 3 about growth factors is not really needed, as they aren't the only cargo that EVs transport. What about lipids, DNA, RNA?

Response 1: Noted, the section has been omitted.

Comments 2: I do not understand the reasoning behind including GF in the review as they can be inside the EV or as a synergic effect. Are there published papers where they combined all of them? Aren't EVs supposed to have an equivalent effect or contain this GF? What would be the complications associated with including EVs and multiple GF, and how the dosing be calculated?

Response 2: Similarly, please refer to the response to Question 1. Thank you.

Comments 3: The images from the figures are not of very good quality (e.g. Figures 3, 4). The authors should include more figures where results from the published papers appear not only resume 3-5 papers in tables 1 and 3. The selection of papers is based on what consideration? Only 3 or 5 papers were published in the field in the past years?

Response 3: Noted, improved figures were re-attached in the manuscript. Alternatively, the original JPEG versions will be uploaded as supplement. Also, we further expanded on the examples described in Section 5.1 and Table 3 [Line 357 – 395]. The search was for the most pertinent representative papers to ensure the compactness of the manuscript.

Comments 4: The tables are not discussed in the text, so I recommend making tables with complete information or changing them by figures where results from papers are presented.

Response 4: Thank you for pointing this out, all tables are now added to the text.

Reviewer 2 Report

Comments and Suggestions for Authors

The manuscript is based on a well-thought-out structure, and its clarity in writing makes it easy to understand the application of bioprinted EVs in skin regeneration. The inclusion of more empirical evidence and practical insight would likely enhance the impact and relevance of the current work.

This review outlines the great promise of combining 3D bioprinting with EVs. However, it would be more effective if it delved deeper into the translational components and real-life applications of this technology.

In conclusion, the paper offers a significant contribution to regenerative medicine by suggesting a novel approach that leverages the synergistic effects of bioprinted EVs. With the addition of more empirical data and practical insights, it has the potential to become a valuable reference for researchers in this field.

Comments on the Quality of English Language

The manuscript demonstrates a solid command of technical language and a structured presentation of complex topics. However, attention to grammar, consistent terminology, and smoother transitions can enhance clarity and readability. By addressing these areas, the overall quality of the manuscript’s English can be significantly improved, making it more accessible to a broader audience.

Author Response

Comments 1: The manuscript is based on a well-thought-out structure, and its clarity in writing makes it easy to understand the application of bioprinted EVs in skin regeneration. The inclusion of more empirical evidence and practical insight would likely enhance the impact and relevance of the current work.

Response 1: Noted, we further expanded on the examples described in Section 5.1 and Table 3 [Line 357 – 395]. The search was for the most pertinent representative papers to ensure the compactness of the manuscript.

Comments 2: This review outlines the great promise of combining 3D bioprinting with EVs. However, it would be more effective if it delved deeper into the translational components and real-life applications of this technology.

Response 2: To our knowledge, only native EVs and 3D bioprints have made progress into clinical trials. However, there is yet to be any clinical trials registered for the application of 3D bioprinted EVs. We have added a few examples of exosome incorporated into hydrogel for skin regeneration with/without 3D bioprinting examples described in Section 5.1 [Line 357 – 395]

Comments 3: In conclusion, the paper offers a significant contribution to regenerative medicine by suggesting a novel approach that leverages the synergistic effects of bioprinted EVs. With the addition of more empirical data and practical insights, it has the potential to become a valuable reference for researchers in this field.

Response 3: Thank you for your insights into improving the manuscript, we hope the corrections will have met your expectations.

Reviewer 3 Report

Comments and Suggestions for Authors

The document “Exploring Synergistic Effects of Bioprinted Extracellular Vesi cles for Skin Regeneration” is well focused and presents outstanding information on the application of extracellular vesicles in 3D printing and regenerative medicine, and also lists some of the factors of growth and its most used functions. It also highlights the importance of the minimum requirements to generate bioinks and their application in 3D printing in the creation of tissue structures including extracellular vesicles. The paper is well referenced and highlights the potential application of vesicle regenerative medicine.

-Some figures need to be improved, they are of low quality.

Comments on the Quality of English Language

no comments

Author Response

Comments 1: Some figures need to be improved, they are of low quality.

Response 1: Noted, improved figures were re-attached in the manuscript. Alternatively, the original JPEG versions will be uploaded as a supplement.